# Selection of tRNA Genes in Human Breast Tumours Varies Substantially between Individuals

**DOI:** 10.3390/cancers15143576

**Published:** 2023-07-12

**Authors:** Sienna P. Butterfield, Rebecca E. Sizer, Emma Rand, Robert J. White

**Affiliations:** Department of Biology, University of York, York YO10 5DD, UKemma.rand@york.ac.uk (E.R.)

**Keywords:** breast cancer, epigenetics, H3K27ac, RNA polymerase III, tRNA

## Abstract

**Simple Summary:**

Elevated expression of many tRNAs is a consistent feature of breast cancer. As each tRNA is encoded by multiple genes, we investigate whether the same genes are always overexpressed in each breast tumour, or whether distinct members of a gene family become activated from one tumour to the next. We find considerable variation between tumours as to which family members undergo activation, whereas familial patterns show much more consistency. A significant correlation is found between patient prognosis and the expression levels of specific tRNAs.

**Abstract:**

Abnormally elevated expression of tRNA is a common feature of breast tumours. Rather than a uniform increase in all tRNAs, some are deregulated more strongly than others. Elevation of particular tRNAs has been associated with poor prognosis for patients, and experimental models have demonstrated the ability of some tRNAs to promote proliferation or metastasis. Each tRNA isoacceptor is encoded redundantly by multiple genes, which are commonly dispersed across several chromosomes. An unanswered question is whether the consistently high expression of a tRNA in a cancer type reflects the consistent activation of the same members of a gene family, or whether different family members are activated from one patient to the next. To address this question, we interrogated ChIP-seq data to determine which tRNA genes were active in individual breast tumours. This revealed that distinct sets of tRNA genes become activated in individual cancers, whereas there is much less variation in the expression patterns of families. Several pathways have been described that are likely to contribute to increases in tRNA gene transcription in breast tumours, but none of these can adequately explain the observed variation in the choice of genes between tumours. Current models may therefore lack at least one level of regulation.

## 1. Introduction

Human genomes are currently estimated to carry 429 genes for tRNAs that are predicted with high confidence to be functional [1]. Of these, around half are expressed in a given cell type [2], which implies considerable redundancy, with multiple loci contributing each of the 46 anticodons used by human cells. For example, four genes are transcribed to give tRNA-Arg-CCG and nine genes contribute to tRNA-iMet-CAU. The tRNAs that share an anticodon are referred to as an isoacceptor family; sometimes, they differ elsewhere in their sequence, in which case the variants are known as isodecoders. In certain instances, distinct isodecoders may differ in function [3], although this remains to be tested in most cases.

All nucleated human cells are assumed to express each of the 46 isoacceptors, but the levels of tRNAs vary according to the conditions. For example, the rate of tRNA synthesis fluctuates during passage through the cell cycle [4]. Although the total level of tRNA increases during proliferation, a subgroup of tRNAs are expressed less strongly under proliferative conditions and instead may be induced during differentiation [5]. Such variations in the relative levels of tRNAs have been postulated to impact the proteomic composition of cells, since the rate at which any codon is translated is responsive to the availability of cognate tRNAs [5]. Direct evidence to support this model came from proteomic analyses after the specific overexpression of a selected tRNA [6]. Thus, raising the level of tRNA-Arg-CCG led to the increased translation of mRNAs enriched in the cognate codon, and the same was true of tRNA-Glu-UUC [6]. As well as stimulating translation, an abundance of tRNAs can also selectively improve the stability of mRNAs enriched in cognate codons [6,7]. A cell’s proteome is therefore sensitive to its tRNA composition. 

Abnormal patterns of tRNA expression have been observed in many types of cancer [8,9,10,11,12]. For example, several studies have reported that tRNAs are aberrantly expressed in breast cancers relative to normal breast tissue [13,14,15,16]. There is an overall increase in the abundance of tRNA in the tumours, but changes are not uniform; certain tRNAs are overexpressed up to 10-fold, whereas others change little or even decrease in cancer cells [13,14,15,16]. Fourteen specific tRNAs were reported to be significantly associated with poor prognosis in patients with breast cancer [14]. Furthermore, a causal role was clearly established for tRNA-Arg-CCG and tRNA-Glu-UUC in promoting invasion and metastasis by breast cancer cell lines in culture and in mouse models [6]. This could be explained mechanistically by the enhanced stability and/or translation of mRNAs enriched in cognate codons that encode pro-metastatic proteins [6]. Thus, tRNA expression patterns can have functional impacts in ways that are relevant to disease pathology and survival. 

The quantification of tRNA expression is problematic because tRNAs are highly modified and have stable secondary structures. Assays based on hybridisation, such as Northern blots and microarrays, have long been considered reliable but are not amenable to high-throughput analyses and cannot distinguish between tRNAs that differ by fewer than eight bases [17], a serious limitation given that examples exist of tRNAs that differ at only a single base yet recognise distinct codons. Furthermore, post-transcriptional modifications found in mature tRNAs can disrupt hybridisation with probes, thereby introducing biases, which might underestimate highly modified forms [18]. As the extent of modification varies between different tRNAs and cell types [17,19], comparisons may be less reliable than is often assumed. Sequencing-based approaches, which are standard for other types of RNA, are severely impeded by the difficulty of producing complementary DNA from tRNAs, because nucleoside modifications in mature tRNAs cause pauses or stops during reverse transcription [20]. To overcome this challenge, a variety of specialised approaches have been developed to facilitate high-throughput sequencing [20]: several examples use a more processive reverse transcriptase, sometimes with the addition of prokaryotic demethylases to remove certain base modifications [19,21,22,23]; another approach, termed Hydro-tRNA-seq, uses limited fragmentation during library preparation to increase the uniformity of coverage by priming shorter tRNA fragments [24]. Each approach provides benefits, but correlations between methods are often poor [19,23], which suggests that imperfections and inherent biases remain. 

An orthogonal approach is to infer expression by using ChIP-seq to determine the presence at tRNA genes of RNA polymerase (pol) III, the enzyme that is uniquely responsible for their transcription. Not only does this avoid the challenges of quantifying transcripts that are highly modified and folded, but it is also able to distinguish between genes encoding identical tRNAs, because ChIP-seq reads can include unique flanking DNA. Balanced against the superior quantification and resolution of ChIP-seq is the fact that it measures a proxy of transcription rather than tRNA levels; the activity of pol III is inferred from its detection, but the approach is blind to any post-transcriptional regulation that might influence tRNA abundance by controlling its stability. Nevertheless, ChIP-seq provides valuable information about which genes attract RNA polymerases and how this might vary between cell types and conditions. To date, a relatively small number of studies have used antibodies against pol III for ChIP-seq (e.g., [25,26,27,28,29,30,31,32]), whereas abundant data are publicly available concerning histone modifications. This resource can be leveraged to provide insight into the status of tRNA genes, as it is known how acetylations and methylations at specific positions of histones correlate with the presence of pol III. For example, the acetylation of histone H3 at lysine 27 (H3K27ac) correlates with tRNA expression, pol III occupancy and chromatin accessibility at tRNA genes in human cells [5,25,26,27,33]. As the same mark is also associated with active pol II promoters and enhancers, it has been widely investigated in contexts that have yet to be tested with pol-III-specific reagents. For example, H3K27ac has been mapped by ChIP-seq in clinical specimens of luminal breast cancer [34]. We have mined these results to compare the statuses of tRNA genes between tumours, inferring gene activity from the strength of H3K27ac. We complemented this approach with tRNA levels detected within miRNA-seq data from The Cancer Genome Atlas (TCGA), a comprehensive resource that includes data from ~10,000 human patients [35]; although miRNA-seq was not designed to measure tRNA, it was reported [15] to provide measurements that correlate well (Spearman’s correlation Rs = 0.73, *p* < 2 × 10^−16^) with results obtained using DM-tRNA-seq, a technique designed specifically to quantitate tRNAs [22]. Neither of the approaches that we have used can be considered optimal in assaying tRNA gene activity, but they provide opportunities to leverage data from patient cohorts that are unlikely to ever be assayed in such numbers using specialised pol-III-specific protocols and reagents. 

## 2. Materials and Methods

### 2.1. Breast Cancer H3K27ac ChIP-seq Data

#### 2.1.1. Downloading ChIP-seq Datasets

H3K27ac ChIP-seq datasets from primary and metastatic ERα-positive luminal breast cancers [34] are available at https://www.ebi.ac.uk/ena (accessed on 20 July 2022) under project number PRJEB22757 in the European Nucleotide Archive (ENA). H3K27ac and matching input ChIP-seq data were downloaded in Binary Alignment Map (BAM) format. 

#### 2.1.2. Regions of Interest

Tracked GRCh38 (hg38) tRNA genes as a complete table were downloaded from UCSC (available at https://genome.ucsc.edu (accessed on 20 July 2022)) using the Table Browser function and converted to a .txt file [36].

#### 2.1.3. Quantification of H3K27ac Signals Using Easeq 

BAM files containing filtered alignments for H3K27ac ChIP-seq and corresponding input files were loaded as ‘Datasets’ into Easeq (available at https://easeq.net (accessed on 20 July 2022)) [37]. tRNA genes from 2.1.2. were imported as ‘Regionsets’. The ‘Quantify’ tool in Easeq was used to quantify H3K27ac peaks at regions of interest. Quantification values or ‘Q-values’ were generated via this function by counting the number of reads in the ‘Dataset’ that overlapped with regions of interest within the ‘Regionset’. These quantification analyses were performed at tRNA genes ±500 bp from the centre of the gene. Default settings of normalising counts to DNA fragments and normalising to reads per million were maintained, whilst the setting of ‘normalise signal to a size of 1000 bp’ was deselected. Q-values were generated at regions of interest for H3K27ac and corresponding input ‘Datasets’ for 29 primary and 11 metastatic tumour samples. 

#### 2.1.4. Data Analysis

Q-values generated in EAseq were exported into Excel and H3K27ac values were normalised to corresponding input values. 

The tRNA gene set from UCSC was modified by removing tRNAs not present in the Genomic tRNA Database high-confidence set of 429 tRNA genes from the hg38 genome (gtRNAdb, https://gtrnadb.ucsc.edu (accessed on 20 July 2022)), except for tRNA-SeC-2-1. This included nuclear-encoded mitochondrial tRNAs (nmt-tRNAs), tRNA-like predictions and two tRNA genes that did not align to a recognised chromosome, leaving 428 tRNA genes for analysis. Data visualisation was performed using ‘Conditional Formatting’ in Excel to create heat maps. Where appropriate, statistical analyses were performed using the Student *t*-test. 

### 2.2. Comparison of tRNA Isoacceptor Expression between Tumour and Normal Breast Tissue

Expression levels of tRNA isoacceptor families were assessed using the codon level analysis module available at the tRiC database (https://hanlab.uth.edu.tRic (accessed on 19 October 2022)) [38]. The expression of selected tRNAs in normal breast (*n* = 104) and invasive breast carcinomas (*n* = 1077) from TCGA (BRCA) samples was compared. 

### 2.3. Analysis of Correlation between tRNA Expression and Patient Survival

Expression and clinical data including survival were available for 990 patients. Kaplan–Meier survival curves were created to compare the survival of those in the highest versus the lowest quartiles for the expression of each isoacceptor family in R [39], using the survival package [40,41]. In addition, Cox proportional hazards regression models were fitted with expression as a continuous predictor. 

## 3. Results

### 3.1. H3K27ac Enrichment Varies Considerably between tRNA Gene Loci and also between Tumours

ChIP-seq data from tumour samples of forty patients with ER+ luminal breast cancer were examined to determine the extent of the lysine 27 acetylation of histone H3 at 428 loci predicted in gtRNAdb to encode functional tRNA. The results are presented in a heat map, where the H3K27ac signal for each tRNA gene is depicted for each tumour by a gradient of colour coding, with red marking the highest acetylation and blue the lowest (Figure 1A). Each column presents data from a different patient’s cancer; 29 are from primary tumours and 11 are from metastases; the latter are grouped on the right side of the map and are separated by a black vertical line from the primary tumours. Each horizontal line represents a different tRNA gene and displays the H3K27ac at that locus in each of the forty cancers. The order of the tRNA genes corresponds to their genomic positions, beginning at chromosome 1. 

It is immediately apparent that the strength of the signal varies between loci; this is expected, as the sequence of the promoter within tRNA genes shows variations that impact activity. Dependent on its sequence, the internal promoter attracts TFIIIC, a transcription factor that is required for pol III recruitment and the transcription of tRNA genes [42]. TFIIIC has been shown to interact with p300, one of the two enzymes responsible for H3K27ac in mammalian cells (the other being its relative CBP) [43]. Accordingly, p300 is detected at large numbers of tRNA genes throughout the human genome [44]. Although some tRNA loci are more consistently enriched for H3K27ac than others, it is very striking that there are no examples where the signal is low in every one of the forty tumours analysed. This can be explained by the fact that conformity with the consensus sequence for active internal promoters is one of the criteria used to predict a functional tRNA gene; apart from the two tRNA-Sec-TCA genes, all the tRNA genes investigated here have promoter sequences that are predicted to recruit TFIIIC, which may then attract p300. Most tRNA genes also recruit MYC [28,45], which in turn can recruit both p300 and CBP [46]. In addition, ERα binds a large number of tRNA genes and is itself bound by p300 and CBP [47]. Each of these loci therefore is likely to have the capacity for the strong acetylation of H3K27, but it is clear from Figure 1A that this potential is not fully realised in these breast tumours and H3K27ac is relatively low at many of the tRNA genes, which is depicted as blue colouration in the heat map. 

The short arm of chromosome 6 contains 157 tRNA genes in a very large cluster from 6p21.2 to 6p22.3 [48]. This region is indicated in the heat map in Figure 1A by a black box, where it is evident that a disproportionately large number of tRNA genes in this region have relatively low H3K27ac in most of the cancers. However, each gene has stronger acetylation in a few of the tumours and several genes in this region have relatively high H3K27ac in the majority of cancers. 

The variation between cancers is striking. Certain features are shared by many examples, but there are always idiosyncrasies, with no two cancers showing identical patterns. There is also inequality between tumours. The heat map in Figure 1B represents, for each cancer, the total H3K27ac signal from the 428 tRNA gene loci investigated. Considerable dissimilarity is evident, with the highest overall signal (tumour G_016) 5.8-fold stronger than the weakest (P_004). Despite the abnormally high overall H3K27ac across the tRNA genes in tumour G_016, there remain large numbers of loci where acetylation is low, just as the overall relative weakness of H3K27ac in P_004 does not preclude a strong signal at many individual tRNA genes. Indeed, there are examples of loci with stronger H3K27ac in P_004 than in G_016, such as the tRNA-iMet-CAT-1-5 locus. Thus, inequalities between cancers in the overall strength of this histone modification do not override local regulatory effects at specific genes.

To quantify the extent of variation between individual tumours, normalised Q-values for H3K27ac at each tRNA gene were compared between pairs of cancers. We compared two primary tumours with very similar overall levels of H3K27ac across the 428 tRNA loci (P_011 and G_018; Figure 2A). This revealed considerable variation at individual genes (Spearman’s rank correlation coefficient *p* = 0.570), with many loci differing substantially in H3K27ac Q-values between the pair of cancers, despite the fact that they were selected because of their close similarity in overall H3K27ac across the set of loci. However, the variation was markedly reduced when comparing the total H3K27ac Q-value for isoacceptor families, each representing the sum of values for the individual genes with the same anticodon (*p* = 0.732; Figure 2B). This higher consistency in isoacceptor activity suggests that compensatory changes amongst family members might mitigate fluctuations at individual loci to reduce variation. This apparent stabilising effect is even stronger at the level of isotypes, comprising all tRNAs that carry the same amino acid (*p* = 0.906; Figure 2C). Comparisons between other tumour pairs show the same pattern, with correlations much higher at the levels of isoacceptors and isotypes than for individual loci (Appendix A). Thus, tRNA gene families appear able to dampen the variability in H3K27ac that occurs at individual loci. This illustrates how gene redundancy might allow greater consistency of transcript expression. 

### 3.2. Comparison of H3K27ac Enrichment at tRNA Gene Loci between Primary and Metastatic Cancers

The metastatic samples analysed are secondary breast cancers from several locations, mainly the liver or lymph nodes. Despite having grown at different sites, they show similar overall patterns of H3K27ac enrichment to the primary tumours. This is illustrated by Figure 3A, which displays the average H3K27ac for all the primaries alongside the average signal for all the metastases at each tRNA gene. There is striking conformity between the two sets, with most loci showing similar H3K27ac in the primary and secondary cancers. However, there are also multiple exceptions, where individual tRNA genes show stronger activation in one set or the other. Thus, the metastatic spread of breast cancers is accompanied by selective changes in the chromatin state of a subset of tRNA genes. 

When all the tRNA loci and cancers are summed, the average strength of H3K27ac is higher in the primaries than in the cancers that have metastasised, although the difference falls short of statistical significance (Figure 3B). If the H3K27ac Q-values are compared between primary and metastatic cancers, it is again apparent that anticodon isoacceptor and amino acid isotype families show less variation than individual tRNA genes (Figure 3C–E). The relative distribution of H3K27ac between the isotype families is very similar between the primary tumours and metastases (Figure 3F).

### 3.3. tRNA-iMet and tRNA-Met Genes

The tRNA-iMet-CAT genes encode the initiator tRNA (tRNA-iMet-CAU) that is required to commence the translation of mRNAs. This unique role is of particular importance, as polypeptide initiation can be rate limiting for translation. When the stable transfection of a tRNA-iMet-CAT gene was used to double the levels of this tRNA in two lines of untransformed breast epithelial cells, proliferation increased significantly [49]. An independent study generated stably transfected derivatives of a breast carcinoma line in which tRNA-iMet-CAT expression was raised ~1.5-fold; this also stimulated proliferation significantly, as well as decreasing apoptosis [18]. In contrast to these proliferative responses in breast epithelial cells, 2.1-fold overexpression of tRNA-iMet-CAT in a laryngeal squamous cell carcinoma cell line suppressed proliferation in favour of apoptosis [50]. Other studies found no effect of tRNA-iMet-CAT overexpression on the proliferation of human glioma cells [51], mouse melanoma [52] or mouse embryonic fibroblasts [53]; however, it had other oncogenic effects in these systems. Thus, tRNA-iMet-CAT stimulated soft agar colony formation when overexpressed in human glioma cells [51], whereas it increased invasive and migratory behaviour when overexpressed in fibroblasts or melanoma cells from mice [52,53]. Because of these reported links with oncogenesis, we focused first on the tRNA-iMet-CAT genes. 

The analysis of 1077 patient samples in the TCGA database found significantly higher levels of tRNA-iMet-CAU in breast invasive carcinomas compared with normal breast tissue (*p* = 6.3 × 10^−6^; Figure 4A). However, patients with tumours in the top quartile of tRNA-iMet-CAU expression did not differ significantly in 5-year survival from patients whose tumour tRNA-iMet-CAU levels fell within the lowest quartile (*p* = 1.0; Figure 4B). The hazard ratio is also not significant when the levels of this tRNA are treated as a continuous variable (*p* = 0.70; HR = 1.05). Thus, patient survival in the TCGA breast cancer cohort does not correlate significantly with the expression of tRNA-iMet-CAU in their tumours.

Eight human tRNA-iMet-CAT genes are identical in sequence (iMet-CAT-1-1 to iMet-CAT-1-8) and are all predicted to be active, based on the sequence features of the genes and their flanking DNA [54]. As these genes have identical binding sites for TFIIIC (within their transcribed regions), they might be expected to recruit p300 with similar efficiencies, leading to similar strengths of H3K27ac. It is therefore noteworthy that H3K27 is in fact much less strongly acetylated at three of these genes (1-4, 1-5 and 1-6) relative to the other family members in most of the tumours (Figure 4C,D). This provides clear evidence that the epigenetic status of a tRNA gene can be strongly influenced by its flanking sequences and/or chromosomal environment. In contrast to the eight tRNA-iMet-CAT-1 genes, the tRNA-iMet-CAT-2-1 gene was predicted to be inactive, based on the DNA sequence [54]. In support of this, H3K27ac is relatively low at this gene in most tumours analysed, although higher acetylation is apparent in some cases. Comparison between samples reveals substantial variation between tumours, with each tRNA-iMet-CAT gene showing a pattern that is distinct from that of all other family members. Differences in the strength of this modification between metastases and primary tumours are not consistent between loci and only reach statistical significance for one family member (Figure 4D).

For comparison with the initiator tRNA-iMet-CAT family, we also analysed the elongator tRNA-Met-CAT family, which delivers methionine to growing polypeptides. Again, this is expressed more strongly in invasive breast carcinomas than in normal breast tissue (*p* = 5.1 × 10^−11^; Figure 5A). As with the initiator tRNA, there is no significant difference in 5-year survival between patients with upper and lower quartile tumour levels of elongator tRNA-Met-CAU (*p* = 0.97; Figure 5B). Levels of this tRNA are also not associated with survival when analysed as a continuous variable (*p* = 0.33; HR = 0.85). The genes encoding elongator tRNA-Met-CAU show more sequence variations than the initiator tRNA-iMet-CAT genes, with seven distinct isodecoders producing tRNAs with minor sequence differences. It is not known whether such variations influence stability or function. Four tRNA-Met-CAT loci display strong H3K27ac in nearly all tumours, whereas the other loci are only weakly acetylated on H3K27 in most of the samples (Figure 5C,D). Although tRNA-Met-CAT-7-1 was predicted to be inactive [54], H3K27 is acetylated at this locus in several of the cancers.

### 3.4. tRNA-Arg-CCG and tRNA-Glu-TTC Genes 

tRNA-Arg-CCG and tRNA-Glu-UUC have been shown to stimulate the invasion and metastasis of breast cancer cell lines, both in culture and in mouse models [6]. These effects were attributed to the preferential translation of mRNAs enriched in their cognate codons, some of which encode proteins that promote metastasis [6]. In the TCGA breast cancer dataset, tRNA-Arg-CCG is expressed significantly more strongly in tumours than in healthy breast tissue (*p* = 5.3 × 10^−9^; Figure 6A). A previous small-scale comparison of primary breast tumours found significantly elevated tRNA-Arg-CCG and tRNA-Glu-UUC in patients who developed metastatic relapses (*n* = 15) relative to those who remained free of disease (*n* = 8) [6]. In the survival data from the much larger TCGA breast cancer analysis, upper quartile expression of tRNA-Arg-CCG in breast tumours is not associated with 5-year survival that is significantly different from lower quartile expressors (*p* = 0.46; Figure 6B). There is also not a significant correlation with patient survival if the levels of this tRNA are treated as a continuous variable (*p* = 0.083; HR = 0.88).

The human genome contains four genes for tRNA-Arg-CCG, all of which are predicted to be active based on their DNA sequence and genomic context [54]. All four were found to be expressed in breast cancer cell lines and their levels were elevated in metastatic derivatives relative to the parental lines, especially tRNA-Arg-CCG-2-1 and tRNA-Arg-CCG-1-3 [6]. In the breast tumours analysed here, H3K27ac is strongest for tRNA-Arg-CCG-2-1 and especially tRNA-Arg-CCG-1-3 but is low for tRNA-Arg-CCG-1-1 in most samples. The average acetylation of H3K27 is lower at these loci in metastases than in primary tumours (Figure 6C,D). We had anticipated that tRNA genes associated with metastasis would show stronger activation (H3K27ac) in the metastatic cancers, but it is also possible that their effects require induction within primary tumours, as a prelude to the subsequent malignant progression and spread of the cancer.

As with tRNA-Arg-CCG, tRNA-Glu-UUC was shown to have metastatic effects in model systems [6], and its expression is significantly elevated in the TCGA breast cancer cohort relative to healthy breast tissue (*p* = 5.4 × 10^−7^; Figure 7A). However, 5-year survival is not significantly different between the upper and lower quartiles of tRNA-Glu-UUC expression (*p* = 0.12; Figure 7B), and treating its levels as a continuous variable does not reveal a significant correlation with the risk of succumbing to the cancer (*p* = 0.09; HR = 0.85). The human genome contains multiple distinct tRNA-Glu-TTC isodecoder genes and it is not known whether they differ in function. At three of these loci, H3K27ac is relatively high in most of the cancers, whereas the other tRNA-Glu-TTC loci show lower acetylation in the majority of cases (Figure 7C). For the three with generally high H3K27ac, acetylation is weaker on average in the metastases than in the primary tumours, but this difference only reaches statistical significance at the tRNA-Glu-TTC-2-1 locus (Figure 7D). In two of the cancers, high H3K27ac is evident at all seven of the high-confidence tRNA-Glu-TTC genes, whereas another cancer has relatively low H3K27ac at all of these genes; such variation illustrates the heterogeneity of cancers at the molecular level.

### 3.5. tRNA-Ser Genes 

As we found no significant correlation between breast cancer survival and the tRNAs that have been implicated experimentally in model systems, we looked for correlations with other isoacceptor families and also with individual tRNA genes. Cox’s proportional hazards model identified 71 individual tRNAs whose expression correlated significantly with 5-year survival in the TCGA breast cancer data (Figure 8A). These did not include any members of the tRNA-iMet-CAU, tRNA-Arg-CCG or tRNA-Glu-UUC families. Twenty of the 71 tRNAs gave HR > 1.0, which means that their elevated expression correlated with reduced 5-year survival, whereas the remainder gave HR < 1.0, meaning that their high expression correlated with a greater likelihood of surviving for 5 years. As the tRNA genes are scattered throughout the genome, random proximity to a cancer driver or suppressor might result in a pattern of co-regulation that correlates with survival. However, fortuitous association of this type is less likely to explain cases where multiple family members have significant and similar hazard ratios, unless the genes are located in a chromosomal cluster. The tRNA group associated with increased hazard (HR > 1.0) is highly enriched in tRNA-Ser and tRNA-Tyr, with 10 and 6 examples, respectively, out of the total 20. By contrast, the group that correlates with decreased hazard (HR < 1.0) contains none of these but instead includes 18 tRNA-Gly, 9 tRNA-Gln and 7 tRNA-His. The recurrent occurrence of these tRNAs in one group or the other, but not both, suggests functional significance in relation to survival. If such a function is mediated through translational decoding, one might expect it to be apparent also at the family level, and this is indeed the case (Figure 8B), although the hazard ratio for tRNA-Tyr does not reach significance (HR = 0.790, *p* = 0.067). Three of the four tRNA-Ser isoacceptor families occupy the top three positions in terms of the Cox proportional hazard ratio (Figure 8B; tRNA-Ser-AGA HR = 1.82, *p* = 4.77 × 10^−4^; tRNA-Ser-UGA HR = 1.69, *p* = 0.003; tRNA-Ser-CGA HR = 1.32, *p* = 0.092). Furthermore, patients with tumours in the top quartile of expression for tRNA-Ser-AGA and tRNA-Ser-UGA had a significantly lower likelihood of 5-year survival than patients whose tumour levels fell within the lowest quartile (Figure 8C; tRNA-Ser-AGA *p* = 0.0017; Figure 8D; tRNA-Ser-UGA *p* = 0.0095). The only other isoacceptor that correlates significantly with 5-year survival in this comparison is tRNA-Tyr-GUA (Figure 8E; *p* = 0.014). Although the high expression of six individual members of the tRNA-Tyr-GUA family correlates with greater risk (Figure 8A), elevated levels of this isoacceptor family collectively correlate with a reduction in risk (Figure 8E); this anomaly can be explained by the fact that none of the high-risk six are strongly expressed, and together they only contribute 16% of the total tRNA-Tyr-GUA in the breast tumours analysed, a contribution that is outweighed by other family members.

If the survival status is plotted beyond five years, for as long as data are available, only one additional isoacceptor becomes significantly correlated with prognosis, tRNA-Ser-CGA (*p* = 0.0014). Independent evidence linking tRNA-Ser with survival comes from a previous study of 104 formalin-fixed paraffin-embedded breast tumours, which found four members of the tRNA-Ser-AGA family and three members of the tRNA-Ser-UGA family in a set of 14 tRNAs that were associated significantly with shorter overall survival (*p* = 0.0008) [14]. No members of the tRNA-Ser-CGA, tRNA-Tyr-GUA, tRNA-iMet-CAU, tRNA-Arg-CCG or tRNA-Glu-UUC families were found to be associated with survival in that study. We therefore focused on tRNA-Ser-AGA and tRNA-Ser-UGA, as the two tRNAs most consistently implicated in breast cancer. 

Although elevated levels of both these isoacceptors are consistently associated with poor prognosis, their expression was lower in tumours than in healthy breast tissue in the TCGA breast cancer dataset (tRNA-Ser-AGA, *p* = 1.3 × 10^−9^
Figure 9A; tRNA-Ser-UGA, *p* = 1 × 10^−12^, Figure 10A). Nine tRNA-Ser-AGA genes were predicted to be active on the basis of their sequence and context [54], but three of these had low H3K27ac in most of the tumours that we analysed (Figure 9B,C). It is striking that this includes the tRNA-Ser-AGA-1-1 gene, which has the highest tRNA score of this isoacceptor family; repression of this gene may contribute substantially to the overall decrease in tRNA-Ser-AGA in cancers relative to healthy breast tissue. Of the nine tRNA-Ser-AGA genes, H3K27ac is strongest at 2-2, 2-5 and 2-6 in the breast cancers analysed, and, at each of these loci, it is less strong in metastases than in primary tumours (Figure 9C). The four family members associated with survival in the previous study [14] are 1-1, 2-1, 3-1 and 4-1, each of which has a significant HR > 1 (Figure 8A) and relatively low H3K27ac in the cancers that we analysed (Figure 9B). None of these four tRNA-Ser-AGA genes show increased expression in the TCGA breast tumours relative to normal breast tissue.

For the tRNA-Ser-UGA isoacceptor family, three of the four genes predicted to be active were included in the set of 14 that were identified previously as associated significantly with overall survival [14]. The exception is tRNA-Ser-TGA-1-1, which has consistently strong H3K27ac in the cancers of our dataset, in contrast to the other family members (Figure 10B,C). It is logical that a consistently active gene might have less prognostic value than others with more variable activity. Although the acetylation of H3K27 at the tRNA-Ser-TGA-1-1 gene is significantly lower in metastatic cancers than primary tumours, it remains above the level found at the three other members of this isoacceptor family. Each of the four active tRNA-Ser-TGA genes is expressed more strongly in healthy breast tissue than in TCGA breast cancers. In summary, decreased expression of tRNA-Ser-AGA and tRNA-Ser-TGA is a general feature of these breast tumours and correlates with improved survival.

## 4. Discussion

This study has investigated three specific isoacceptors that have previously been implicated in breast cancer progression through experiments conducted in cultured cells and mouse models, tRNA-iMet-CAU, tRNA-Arg-CCG and tRNA-Glu-UUC [6,18,49]. Microarray analysis demonstrated elevated levels of these tRNAs in three ER+ and six ER- breast tumours, relative to three normal breast samples [13]. Our analysis of a much larger cohort provides additional evidence for the consistent and significant elevation of tRNA-iMet-CAU, tRNA-Arg-CCG and tRNA-Glu-UUC in breast cancers. Such increases are not restricted to these specific isoacceptors but appear to be shared by the majority of tRNAs, to varying degrees [13,14,15,16]. A range of established molecular mechanisms are expected to contribute to this global increase in tRNA levels. For example, ERα drives ~75% of breast cancers and includes tRNA genes amongst its many direct targets for transcriptional activation [47,55,56]. MYC is induced by ERα and in turn stimulates tRNA synthesis [45,57]. MYC also raises the expression of POLR3G [32], a pol III subunit that promotes tumour growth and metastasis in triple-negative breast cancer [58]. Other oncogenes that can raise tRNA expression include MDM2, CCND and PIK3CA [59,60,61,62], all of which are sometimes amplified in breast cancers [63]. Conversely, the deletion or mutation of TP53, RB1, BRCA1 and PTEN is common in breast tumours [63] and can also result in increased tRNA expression [61,62,64,65,66,67,68,69]. All of the above regulators have been shown to influence tRNA synthesis broadly through direct or indirect interactions with TFIIIB, a pol-III-specific transcription factor [45,56,59,60,62,65,68,69,70,71,72,73,74]. Although these mechanisms can account for the overall increase in tRNA found in breast cancers, they cannot explain why some tRNA genes respond more strongly than others, because TFIIIB is required in every case. 

A tRNA-selective mechanism involves the micro-RNA mir-34a, a well-characterised tumour suppressor that is induced by TP53 [75]. In addition to several mRNAs that encode oncogenic products, mir-34a was shown to bind precursor tRNA-iMet-CAU, causing its degradation [18]. Inactivation of TP53, a common event in breast cancers, may reduce mir-34a expression and thereby increase the stability of tRNA-iMet-CAU, perhaps boosting its levels relative to other tRNAs; this differential control, specific to one isoacceptor, would be additional to the general increase in tRNA gene transcription that accompanies the deletion or mutation of TP53 [60,65]. 

Despite the many documented mechanisms that can stimulate tRNA synthesis in breast tumours, the expression of tRNA-Ser-AGA and tRNA-Ser-UGA is lower in the cancers than in normal breast tissue (Figure 9A and Figure 10A). A potential explanation comes from evidence that pol III can be bound and inhibited by the progesterone receptor, which is induced by ERα [76]. However, it is unclear why the repressive effect of the progesterone receptor would outweigh, specifically at these genes, the many stimulatory mechanisms that promote the transcription of most other tRNA genes in breast cancers. SOX4 provides an example of a transcription factor that has been shown to repress a subset of tRNA genes selectively [77]. Thus, when SOX4 was overexpressed in glioblastoma cells, it was found to bind 126 tRNA genes, including all eight members of the tRNA-iMet-CAT-1 isodecoder family; although the binding of TFIIIC was unchanged at these genes, that of TFIIIB and pol III was diminished, and reduced expression was observed [77]. Whether SOX4 can also inhibit tRNA production in breast cells was not tested, but it is commonly overexpressed in breast cancer, where it can stimulate invasion, and it correlates significantly with parameters of poor prognosis [78].

None of the documented mechanisms above can explain the heterogeneity observed between tumours as to which individual tRNA genes within an isoacceptor family are active, as judged by the acetylation of H3K27. In actively growing budding or fission yeast, virtually all functional tRNA genes are occupied by pol III [79,80,81,82]. In contrast, only around half are pol-III-bound in human cell lines [25,26,27,28,83] or tissue from a range of mammalian species [29]. Our data are broadly consistent with these findings, if we assume that H3K27ac marks the sites of pol III binding, an assumption that is warranted by previous studies [5,25,26,27,33]. Thus, in the heat maps for the six isoacceptors that we have investigated, approximately half the loci are relatively active and the other half relatively inactive in the majority of tumours. Occasional exceptions can be found; for example, all 13 tRNA-Glu-TTC genes are active in one particular metastasis (M_013, Figure 7C), but this anomaly does not reflect the general hyperacetylation of tRNA loci in this sample, as only three out of nine tRNA-iMet-CAT genes are active in the same cancer (Figure 4C). The most striking feature of the heat maps is that every tumour shows a different pattern of H3K27ac across the tRNA genes. Only one gene has acetylated H3K27 in all 40 cancers analysed (tRNA-Glu-TTC-1-2) and no gene has unacetylated H3K27 in every cancer. tRNA-iMet-CAT-1-4 and tRNA-iMet-CAT-1-6 are identical in sequence, but the former is active and the latter inactive in tumour P_004, whereas the converse is found in tumour G_002 (Figure 4C). Perhaps the local environment of each tRNA gene varies between individual cases of a particular cancer type in a manner that impacts its epigenetic status. There is much yet to learn about the mechanism(s) operating at this level. 

Kutter et al. (2011) used pol III ChIP-seq to compare the use of tRNA genes in different organs and species [29]. A comparison of pol III occupancy in livers from a pair of mice revealed a Spearman’s rank correlation coefficient *p* = 0.92, which is substantially higher than the *p* = 0.69 obtained when we compared H3K27ac at the tRNA genes of two human luminal breast cancers. The high consistency between murine livers reflects the facts that this organ is primarily composed of a single cell type, hepatocytes, and that laboratory mice share a genetic background, diet and environment, none of which apply to clinical samples, especially tumours that contain a range of disparate cell types. When pol III occupancy at mouse tRNA genes was compared between different organs, the liver and testis, the correlation fell to *p* = 0.77; however, high binding consistency was maintained at the level of the isoacceptor and isotype families, with *p* = 0.94 and 0.98, respectively [29]. Similar effects were found when pol III occupancy was compared between mouse liver and muscle, with *p* = 0.87 for tRNA genes, 0.96 for isoacceptors and 0.98 for isotypes [29]. There is thus great consistency between organs in the relative use of tRNA families, despite variation in the selection of individual genes. The same phenomenon was observed when tRNA occupancy by pol III in the liver was compared between six mammalian species; although separated by 180 million years of evolution, human and opossum livers show Spearman’s rank correlation coefficients *p* = 0.67 at the isoacceptor level and *p* = 0.89 at the isotype level [29]. These striking observations provided an unexpected suggestion of regulatory cross-talk between the tRNA genes within families to allow coordinated transcriptional activity. This was supported by the sequencing of tRNA from several human tissues, which revealed that isoacceptor expression does not vary drastically, despite much larger fluctuations in the levels of constituent isodecoders [23]. Our data illustrate this phenomenon further, with isoacceptor and isotype gene families maintaining overall activity that is relatively consistent between tumours, despite substantial variation in the activation states of individual tRNA genes. This familial conservation is robust to the genetic and epigenetic heterogeneity of cancers. 

Although two studies have found that the overexpression of tRNA-iMet-CAU can stimulate the proliferation of breast cell lines [18,49], we found no significant correlation between the levels of this tRNA and the survival of patients with breast cancer. This was also the case for individual members of the tRNA-iMet-CAU family (Appendix A). In addition, we investigated survival in the TCGA melanoma dataset, as tRNA-iMet-CAT transgenes were shown to promote the metastasis of melanoma cells in mice [52]. However, the hazard ratio for tRNA-iMet-CAU in the TCGA melanoma cohort is again insignificant (*p* = 0.70; HR = 1.05). This does not contradict the experimental evidence for the oncogenic effects of tRNA-iMet-CAU, but suggests that any impact on the survival of patients with these types of cancer is insignificant relative to stronger influences. The same is true of tRNA-Arg-CCG and tRNA-Glu-UUC, which can stimulate the invasion and metastasis of breast cancer cell lines in culture and in mice [6], but do not correlate significantly with patient survival in the TCGA breast cancer cohort. In contrast, the tRNA-Ser-AGA and tRNA-Ser-UGA isoacceptor families have high and significant hazard ratios in the TCGA breast cancer cohort, as do nine of their genes. Furthermore, seven individual members of these families were also found to be significantly associated with risk in an independent breast cancer cohort, albeit much smaller [14]. The fact that these tRNAs correlate strongly with poor prognosis in independent cohorts suggests a role in promoting breast cancer, although it does not prove causality. Experimental tests in model systems would be valuable to interrogate the mechanisms underlying these epidemiological correlations. An important first step came from the demonstration that tRNA-Ser-AGA-2-1 promotes proliferation and anchorage-independent colony formation in vitro and tumour growth in mice, when overexpressed ~1.6-fold in a cell line derived from a near-normal bronchial epithelium [84]. The overexpression used in this model mimics the increase in tRNA-Ser-AGA-2-1 found in lung adenocarcinomas in the TCGA database. In contrast, this tRNA decreases in TCGA breast cancers, as do the tRNA-Ser-AGA and tRNA-Ser-UGA families collectively, irrespective of stage, ER or HER2 status. Thus, the changes in tRNA expression that accompany tumourigenesis can vary according to the tissue type. It is noteworthy that breast tumours with the most tRNA-Ser-AGA and/or tRNA-Ser-UGA have the worst prognosis, even though their levels may have fallen relative to adjacent normal tissue. 

This study has made methodological compromises in order to take advantage of powerful datasets. As explained in the Introduction, the quantification of tRNA is extremely challenging and multiple procedures have been used, often yielding discrepancies. The miRNA-seq data that we have interrogated were not designed to assay tRNA but have been reported to show a good correlation in this regard with measurements made using the bespoke DM-tRNA-seq technique [15]. As a proxy for gene activity, we have used ChIP-seq measurements of H3K27ac, although the applicability of this proxy has yet to be confirmed in breast tumours. Pol III occupancy and tRNA expression have been shown to correlate well with H3K27ac in multiple human cell lines and clinical samples of several cancer types [5,25,26,27,33], but breast cancer was not included in the previous studies. It is also regrettable that the ChIP-seq data did not include normal breast tissue for comparison. We acknowledge these limitations and recommend that they are remembered when considering our conclusions. 

## 5. Conclusions

As far as we are aware, our study is the first to address tRNA gene selection in human tumours. Previous work has shown an overall increase in tRNA expression in breast cancer and multiple studies have identified molecular mechanisms that can explain this general elevation. However, relatively little attention has been given to the selective down-regulation of specific tRNA families in tumours, as found for the tRNA-Ser-AGA and tRNA-Ser-UGA isoacceptors in luminal breast cancers. The main novelty of our data is in the discovery that there is striking heterogeneity from one tumour to the next with regard to which members of redundant gene families are active. For many isoacceptor families, this will change the relative levels of isodecoders that share an anticodon but vary elsewhere in their sequences; the functional significance of such changes remains to be explored. Current models to explain how tRNA gene transcription is regulated have focused overwhelmingly on the general machinery that participates in the synthesis of every tRNA. Very little is known about the selection process that determines which individual genes are activated in a given cell or tissue. Clearly, some tRNA genes have optimal sequences that dictate their preferential activation, but our data reveal many instances in which a particular tumour fails to activate a gene that is utilised in most other cases, but instead induces another family member that is relatively inactive in the majority of samples. What determines the abnormal repression of the gene that is usually active? What causes the rarely used gene to be induced in this case? Are these two events coordinated and, if so, how? Such micromanaging of tRNA expression remains largely unexplored territory that might eventually identify principles that can be applied to more complex gene families. 

## Figures and Tables

**Figure 1 cancers-15-03576-f001:**
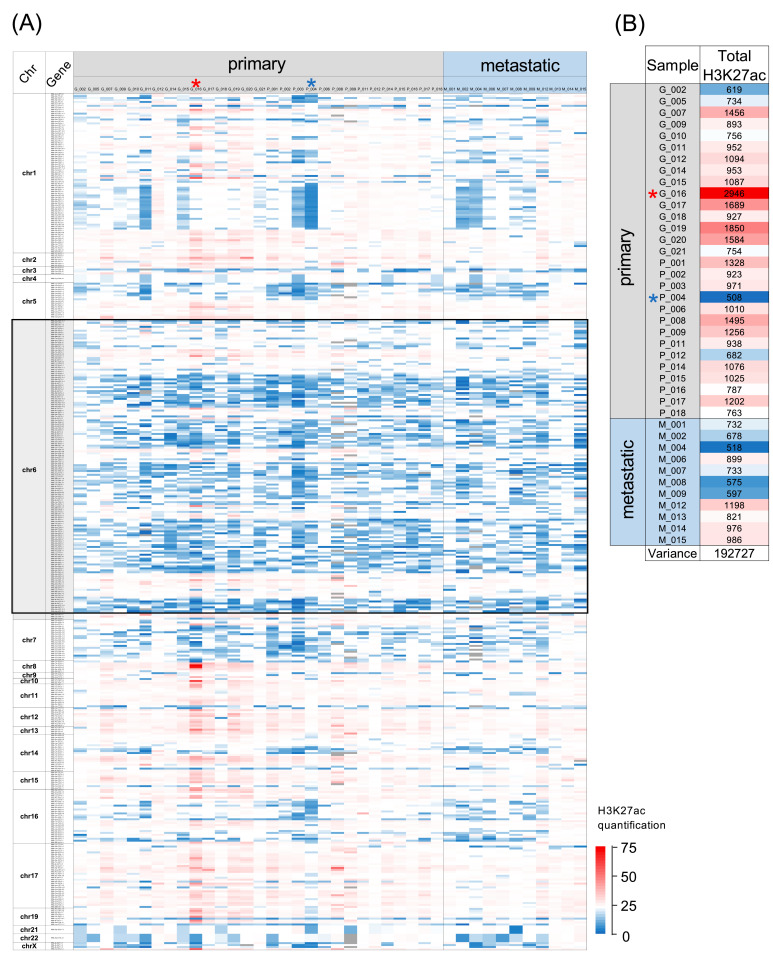
H3K27ac enrichment at tRNA gene loci in ER+ luminal breast cancers. (**A**) Heat map displaying relative H3K27ac signal at 428 tRNA genes in primary (left, *n* = 29) and metastatic (right, *n* = 11) luminal breast cancers. Black box highlights tRNAs on chromosome 6 (chr6). Colour bars denote increasing H3K27ac from blue to red. Grey indicates inadequate data quality. Tumours with the highest and lowest total H3K27ac summed over 428 tRNA genes are marked with red and blue asterisks, respectively. (**B**) Total relative H3K27ac signal in each tumour at 428 tRNA genes. Tumours with the highest and lowest total H3K27ac summed over 428 tRNA genes are marked with red and blue asterisks, respectively.

**Figure 2 cancers-15-03576-f002:**
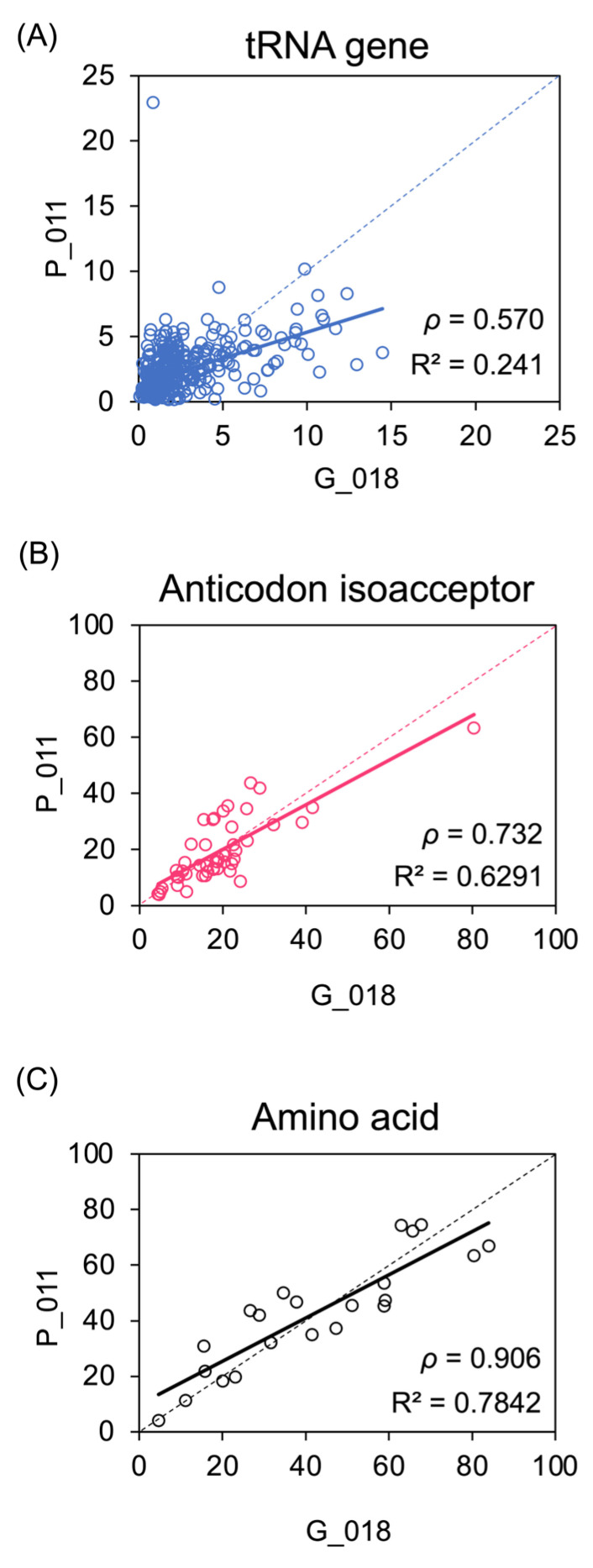
Variation between cancers in H3K27ac enrichment at tRNA genes and families. (**A**–**C**) Plots comparing H3K27ac Q-values at 428 tRNA gene loci (**A**), 48 anticodon isoacceptor families (**B**) and 21 amino acid isotype families (**C**) in tumours P_011 and G_018, which were chosen because they share very similar overall H3K27ac enrichment across the tRNA genes. Spearman’s rank correlation coefficient is shown (*p*).

**Figure 3 cancers-15-03576-f003:**
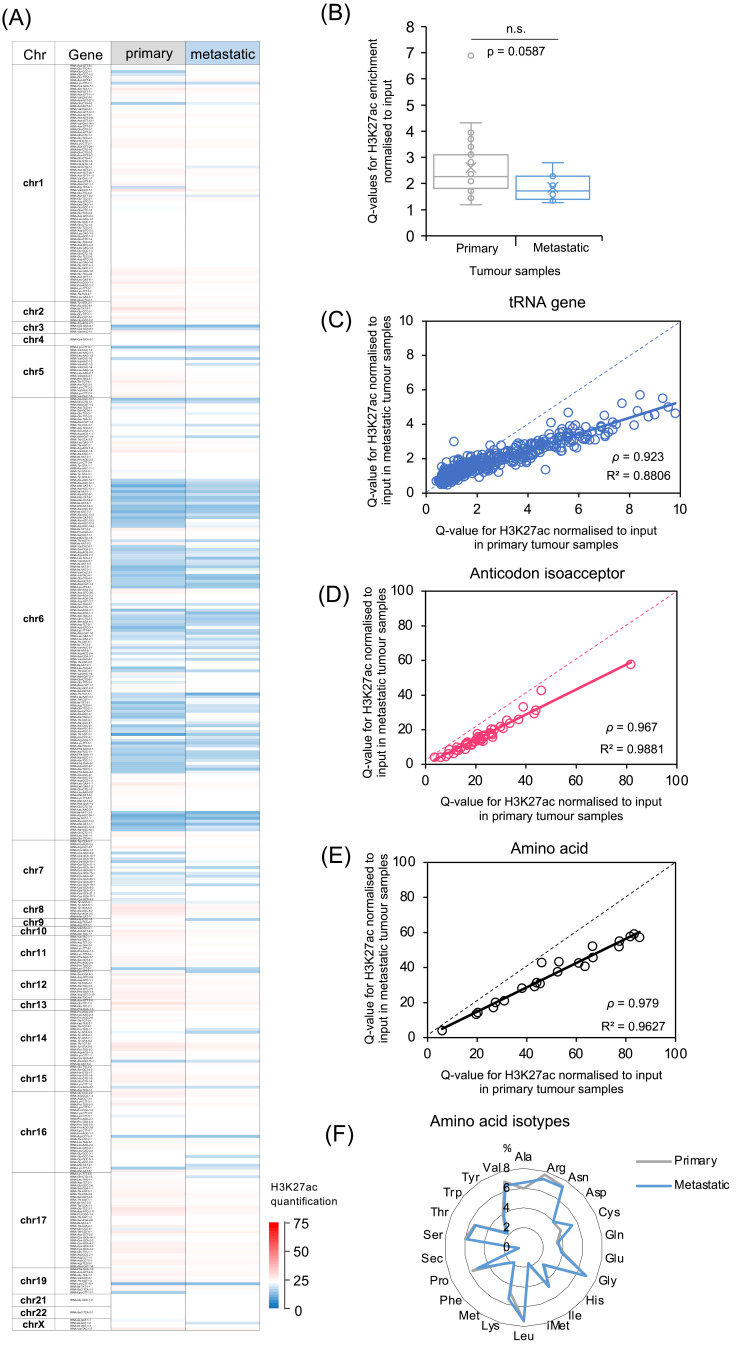
Comparison between primary and metastatic breast cancers of H3K27ac enrichment at tRNA gene loci. (**A**) Heat map displaying relative H3K27ac signal from 428 tRNA genes averaged across primary (left, *n* = 29) and metastatic (right, *n* = 11) luminal breast cancers. (**B**) Relative H3K27ac signal averaged across tRNA genes (*n* = 428) in primary (grey, *n* = 29) vs. metastatic (blue, *n* = 11) luminal breast cancers. Boxes show the median (solid line) ± one quartile, with the mean denoted by a cross; whiskers extend to the furthest data point within 1.5× interquartile range from the box. Student’s *t*-tests have been applied to calculate statistical significance. (**C**–**E**) Plots comparing H3K27ac Q-values in primary vs. metastatic cancers at 428 tRNA gene loci (**C**), 48 anticodon isoacceptor families (**D**) and 21 amino acid isotype families (**E**). Spearman’s rank correlation coefficient is shown (*p*). (**F**) Proportional strength of H3K27ac Q-values at 21 isotypes in primary vs. metastatic cancers.

**Figure 4 cancers-15-03576-f004:**
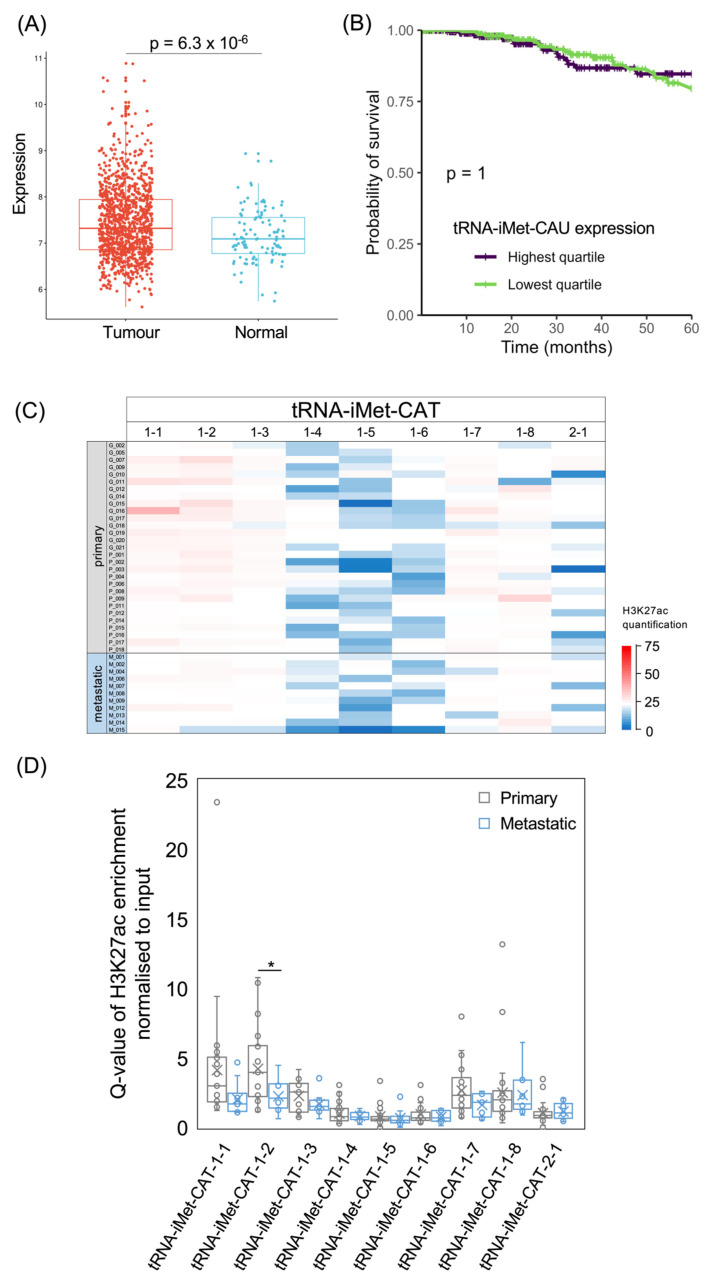
The tRNA-iMet-CAT isoacceptor gene family. (**A**) Relative expression of tRNA-iMet-CAU in normal breast (blue, *n* = 104) and breast invasive carcinoma (red, *n* = 1077). (**B**) Survival of patients with breast invasive carcinomas expressing upper quartile (purple) or lower quartile (green) levels of tRNA-iMet-CAU. (**C**) Heat map displaying H3K27ac signal at tRNA-iMet-CAT genes in primary (upper, *n* = 29) and metastatic (lower, *n* = 11) luminal breast cancers. Colour bar denotes increasing H3K27ac from blue to red. Grey indicates inadequate data quality. (**D**) Relative H3K27ac signal at each tRNA-iMet-CAT gene in primary (grey, *n* = 29) vs. metastatic (blue, *n* = 11) cancers. Boxes show the median (solid line) ± one quartile, with the mean denoted by a cross; whiskers extend to the furthest data point within 1.5× interquartile range from the box. Student’s *t*-tests have been applied to calculate statistical significance, denoted by the *p* value (*p* < 0.05 indicates significance) or * (*p* < 0.05).

**Figure 5 cancers-15-03576-f005:**
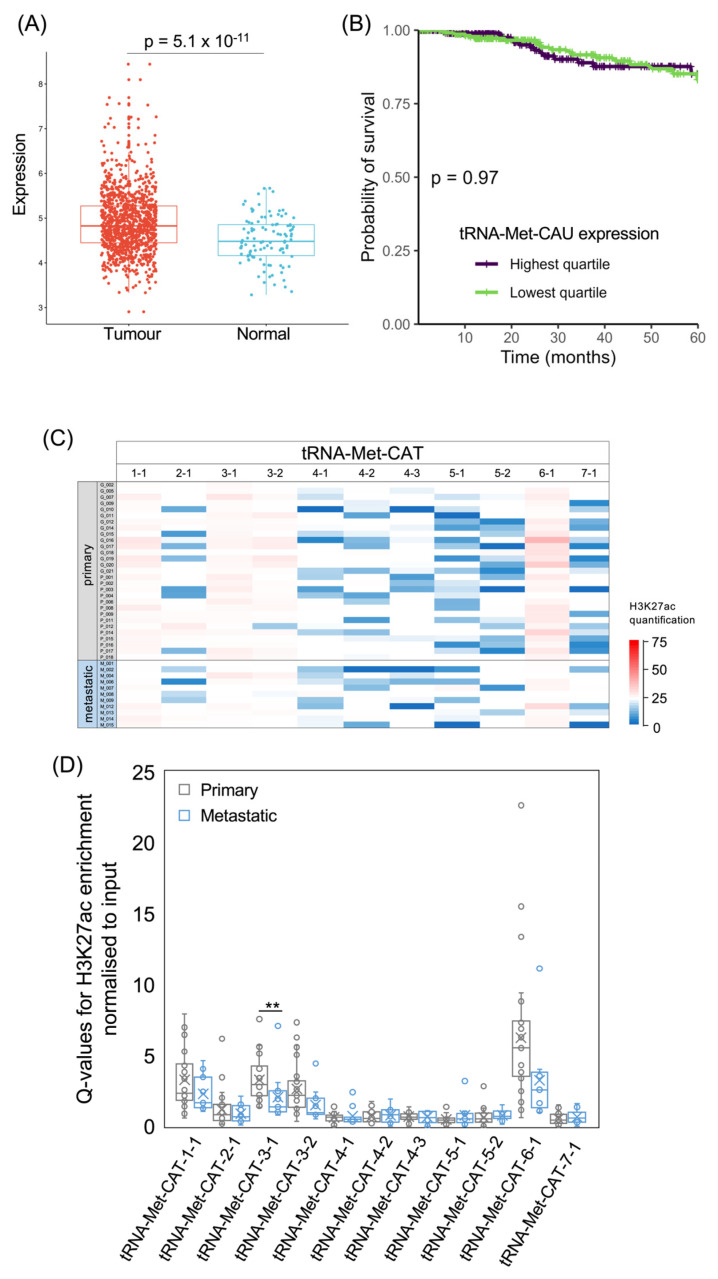
The tRNA-Met-CAT isoacceptor family. (**A**) Relative expression of tRNA-Met-CAU in normal breast (blue, *n* = 104) and breast invasive carcinoma (red, *n* = 1077). (**B**) Survival of patients with breast invasive carcinomas expressing upper quartile (purple) or lower quartile (green) levels of tRNA-Met-CAU. (**C**) Heat map displaying H3K27ac signal at tRNA-Met-CAT genes in primary (upper, *n* = 29) and metastatic (lower, *n* = 11) luminal breast cancers. Colour bar denotes increasing H3K27ac from blue to red. Grey indicates inadequate data quality. (**D**) Relative H3K27ac signal at each tRNA-Met-CAT gene in primary (grey, *n* = 29) vs. metastatic (blue, *n* = 11) cancers. Boxes show the median (solid line) ± one quartile, with the mean denoted by a cross; whiskers extend to the furthest data point within 1.5× interquartile range from the box. Student’s *t*-tests have been applied to calculate statistical significance, denoted by the *p* value (*p* < 0.05 indicates significance) or ** (*p* < 0.01).

**Figure 6 cancers-15-03576-f006:**
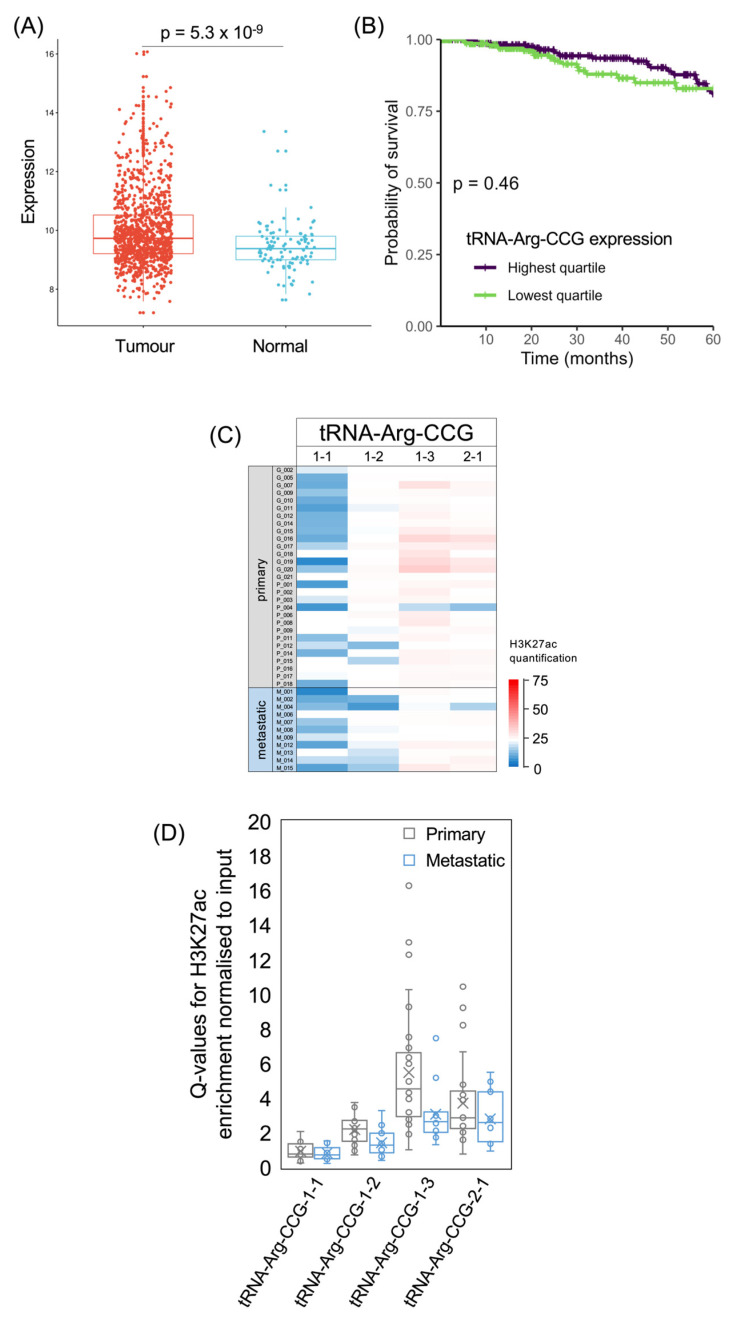
The tRNA-Arg-CCG isoacceptor family. (**A**) Relative expression of tRNA-Arg-CCG in normal breast (blue, *n* = 104) and breast invasive carcinoma (red, *n* = 1077). (**B**) Survival of patients with breast invasive carcinomas expressing upper quartile (purple) or lower quartile (green) levels of tRNA-Arg-CCG. (**C**) Heat map displaying H3K27ac signal at tRNA-Arg-CCG genes in primary (upper, *n* = 29) and metastatic (lower, *n* = 11) luminal breast cancers. Colour bar denotes increasing H3K27ac from blue to red. Grey indicates inadequate data quality. (**D**) Relative H3K27ac signal at each tRNA-Arg-CCG gene in primary (grey, *n* = 29) vs. metastatic (blue, *n* = 11) cancers. Boxes show the median (solid line) ± one quartile, with the mean denoted by a cross; whiskers extend to the furthest data point within 1.5× interquartile range from the box. Student’s *t*-tests have been applied to calculate statistical significance, denoted by the *p* value (*p* < 0.05 indicates significance).

**Figure 7 cancers-15-03576-f007:**
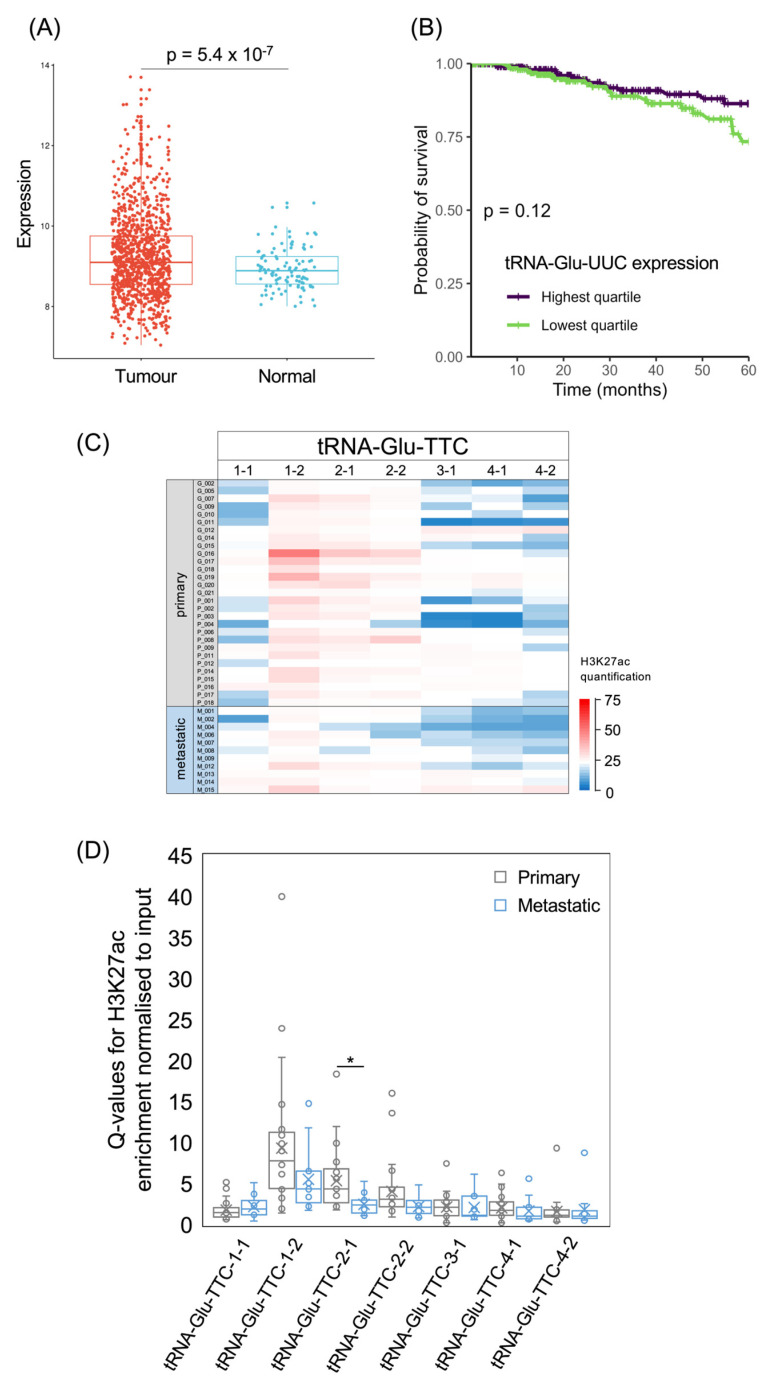
The tRNA-Glu-TTC isoacceptor family. (**A**) Relative expression of tRNA-Glu-UUC in normal breast (blue) and breast invasive carcinoma (red). (**B**) Survival of patients with breast invasive carcinomas expressing upper quartile (purple) or lower quartile (green) levels of tRNA-Glu-UUC. (**C**) Heat map displaying H3K27ac signal at tRNA-Glu-TTC genes in primary (upper, *n* = 29) and metastatic (lower, *n* = 11) luminal breast cancers. Colour bar denotes increasing H3K27ac from blue to red. (**D**) Relative H3K27ac signal at each tRNA-Glu-TTC gene in primary (grey, *n* = 29) vs. metastatic (blue, *n* = 11) cancers. Boxes show the median (solid line) ± one quartile, with the mean denoted by a cross; whiskers extend to the furthest data point within 1.5× interquartile range from the box. Student’s *t*-tests have been applied to calculate statistical significance, denoted by the *p* value (*p* < 0.05 indicates significance) or * (*p* < 0.05).

**Figure 8 cancers-15-03576-f008:**
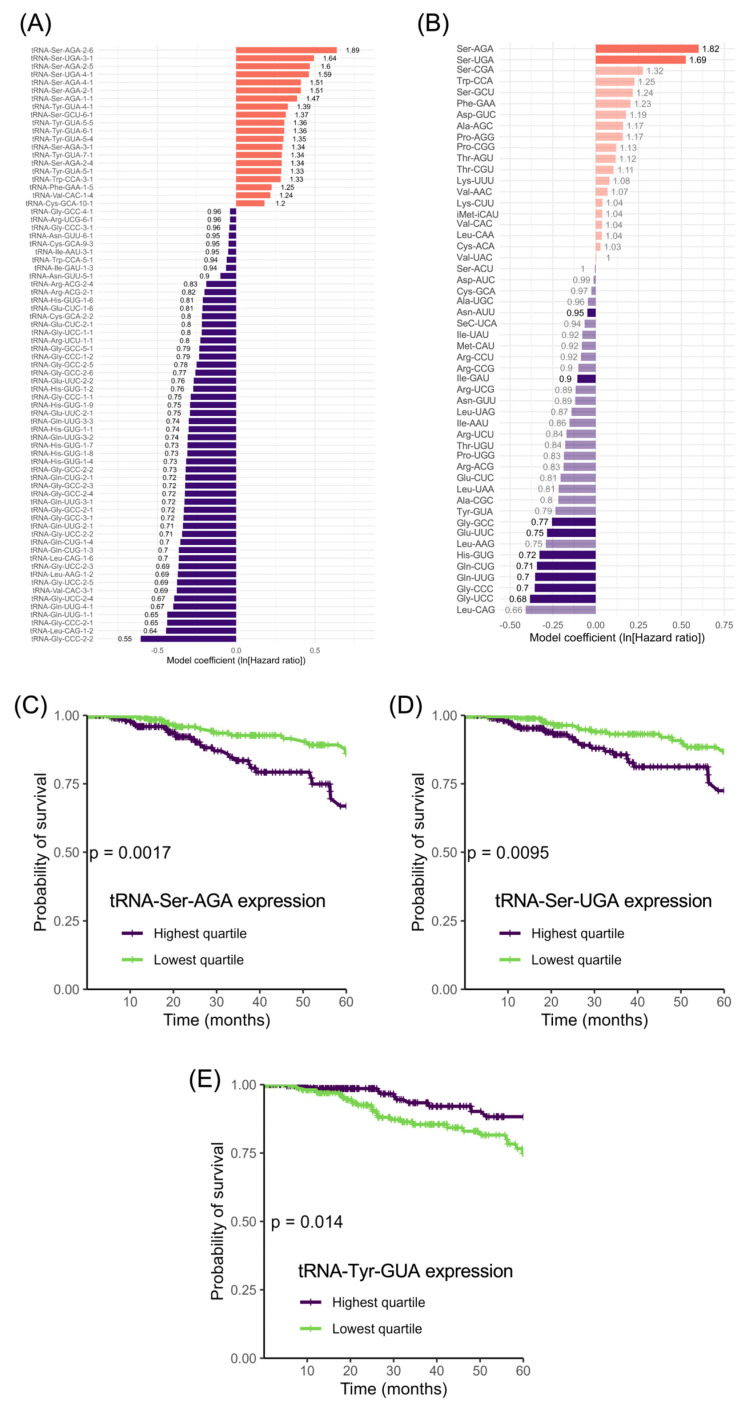
Identification of tRNAs and isoacceptors with expression levels in breast invasive carcinomas that correlate with 5-year survival. (**A**) All tRNAs for which expression levels in breast invasive carcinomas correlate significantly with 5-year survival by Cox proportional hazards regression, ordered from highest to lowest hazard ratio (HR). Red bars indicate HR > 1.0; blue bars indicate HR < 1.0. HR is indicated for each tRNA. (**B**) HR determined by Cox proportional hazards regression for expression levels of each isoacceptor family in breast invasive carcinomas, ordered from highest to lowest HR. Red bars indicate HR > 1.0; blue bars indicate HR < 1.0; shaded bars are not significant. (**C**) Survival of patients with breast invasive carcinomas expressing upper quartile (purple) or lower quartile (green) levels of tRNA-Ser-AGA. (**D**) Survival of patients with breast invasive carcinomas expressing upper quartile (purple) or lower quartile (green) levels of tRNA-Ser-UGA. (**E**) Survival of patients with breast invasive carcinomas expressing upper quartile (purple) or lower quartile (green) levels of tRNA-Tyr-GUA.

**Figure 9 cancers-15-03576-f009:**
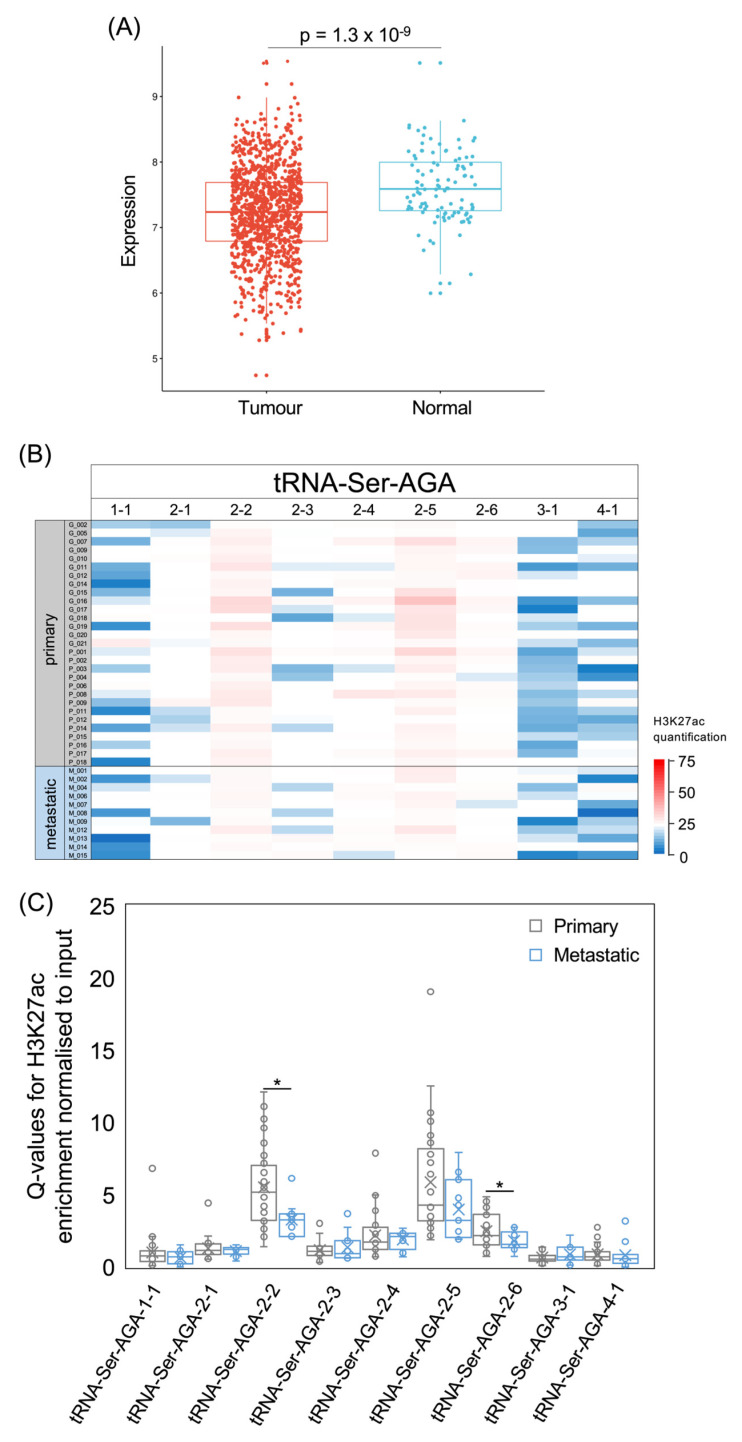
The tRNA-Ser-AGA isoacceptor family. (**A**) Relative expression of tRNA-Ser-AGA in normal breast (blue) and breast invasive carcinoma (red). (**B**) Heat map displaying H3K27ac signal at tRNA-Ser-AGA genes in primary (upper, *n* = 29) and metastatic (lower, *n* = 11) luminal breast cancers. Colour bar denotes increasing H3K27ac from blue to red. Grey indicates inadequate data quality. (**C**) Relative H3K27ac signal at each tRNA-Ser-AGA gene in primary (grey, *n* = 29) vs. metastatic (blue, *n* = 11) cancers. Boxes show the median (solid line) ± one quartile, with the mean denoted by a cross; whiskers extend to the furthest data point within 1.5× interquartile range from the box. Student’s *t*-tests have been applied to calculate statistical significance, denoted by the *p* value (*p* < 0.05 indicates significance) or * (*p* < 0.05).

**Figure 10 cancers-15-03576-f010:**
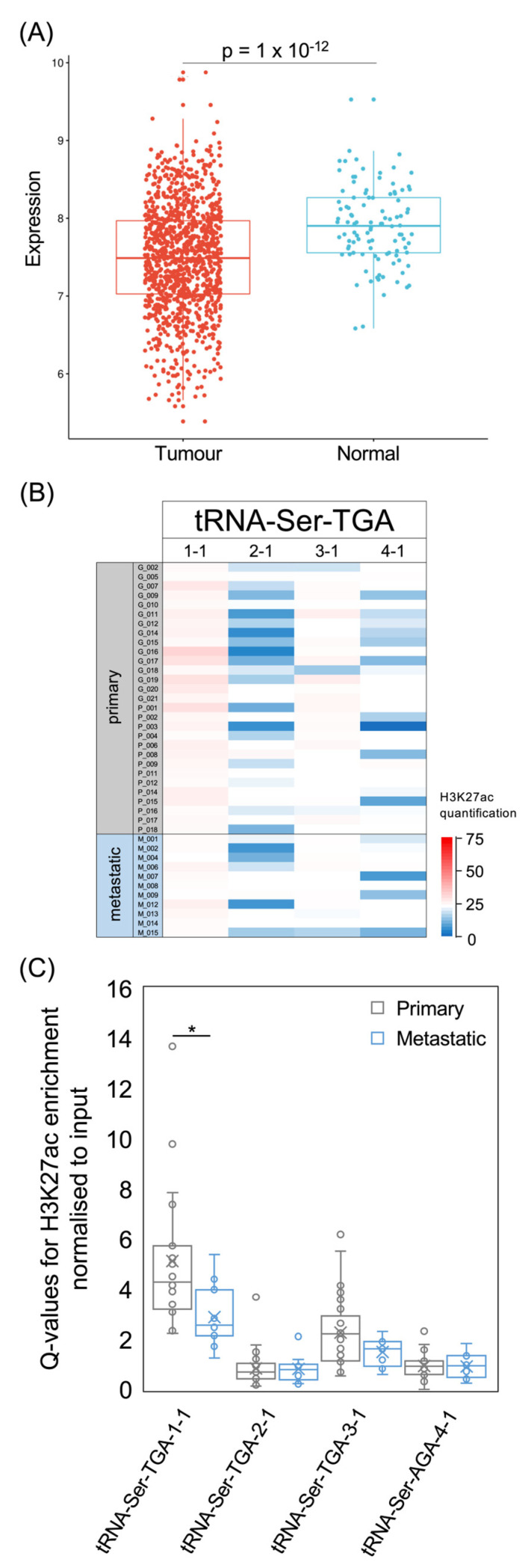
The tRNA-Ser-TGA isoacceptor family. (**A**) Relative expression of tRNA-Ser-TGA in normal breast (blue) and breast invasive carcinoma (red). (**B**) Heat map displaying H3K27ac signal at tRNA-Ser-TGA genes in primary (upper, *n* = 29) and metastatic (lower, *n* = 11) luminal breast cancers. Colour bar denotes increasing H3K27ac from blue to red. (**C**) Relative H3K27ac signal at each tRNA-Ser-TGA gene in primary (grey, *n* = 29) vs. metastatic (blue, *n* = 11) cancers. Boxes show the median (solid line) ± one quartile, with the mean denoted by a cross; whiskers extend to the furthest data point within 1.5× interquartile range from the box. Student’s *t*-tests have been applied to calculate statistical significance, denoted by the *p* value (*p* < 0.05 indicates significance) or * (*p* < 0.05).

## Data Availability

The data can be shared upon request.

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
