# Peer review of "Selection of tRNA Genes in Human Breast Tumours Varies Substantially between Individuals"

_cancers, 2023, doi:10.3390/cancers15143576_

Round 1

Reviewer 1 Report

Increased expression of total tRNAs is commonly observed in cancer but overexpression of specific tRNAs is associated with poor prognosis. Each tRNA is encoded by multiple genes and it is not known if all genes are overexpressed or if only a particular set of genes are causing the increase in expression. To address this, the authors used publicly available H3K27ac ChIP-seq data sets to identify the genes that are actively being transcribed.

A major limitation of this study is the use of a H3K27ac to identify the actively transcribed genes. Ideally ChIP-seq for RNA polymerase (Pol) III specific subunits should be performed. This limitation is clearly started in the manuscript and a valid explanation for the use of a surrogate measure is given by the authors.

Since H3K27ac is also a marker of actively transcribed Pol I and Pol II genes, the authors focus their analysis solely on genome regions encoding tRNA genes. The authors also compare the ChIP-seq results with TCGA miRNA-seq data (as an approximation for tRNA-seq).  Again, although not ideal, the use of the miRNA-seq is clearly explained by the authors.

The data presented in this manuscript is novel and of interest for the broader cancer biology field and specifically the PolII transcription field. The data is well presented and the limitation of the analysis are clearly stated.

Major comments:

1.     Throughout the manuscript comparison are being made between primary and metastatic tumours but there is no reference to the normal levels of H3K27ac over the tRNA genes in heathy tissue. I appreciate that healthy tissue is difficult to obtain and their will be large variation between individuals. However, having some data from heathy tissue (at least in Figure 1A) would make it easier to evaluate the level of H3K27ac deregulation in the tumour samples, specially considering the levels are mostly low or moderate (red and yellow/orange).

2.     In figure 2A a comparison between individual tumours (P_004 and G_16) with similar levels of H3K27ac is shown. The authors state that “Comparisons between other tumour pairs show the same pattern” (ln230). The data for the other tumour analysed should be added to the manuscript as supplementary data.

3.     In section 3.7 the authors conclude that “tRNA genes in the chromosome 1, PER1 and TMEM107 clusters often share an epigenetic state, but this is not apparent at the ALOXE3 cluster” (ln552-553). However, this is solely based on H3K4ac over genes that were previous identified as having barrier activity, by having high H3K27ac at similar level to genes known to have long-range interactions with this cluster.  This is all indirect evidence. To analyse barrier activity the authors should look at histone marks like H3K9me3 and H3k27me3 in the regions adjacent to this clusters. Alternatively, this section could be removed from the manuscript since, has it stands, doesn’t bring much extra information for the reader.

Minor comments:

1.     The green in the heatmaps is very difficult to see. I would suggest changing it to another colour with higher contrast to yellow.

2.     In the introduction the authors state that “To date, a relatively small number of studies have used antibodies against pol III for ChIP-seq” (ln 92) but they do not provide a reference in the text at this point.

Author Response

Reviewer 1 described the manuscript as novel and of interest for the broader cancer biology field, with well-presented data. We are grateful for these comments and for the suggestions for improvement.

Major issues.

  1. The reviewer suggested that it would be informative to include healthy breast tissue as a reference to the normal levels of H3K27ac at tRNA genes. We agree, but such data is not available. Healthy tissue was not included in the dataset that we used for our analysis and we have not found alternative studies that included H3K27ac ChIP-seq data in matched normal and tumour breast samples. Such comparisons will therefore have to await publication of new clinical analyses. A comment regretting the unavailability of such data has been added to our manuscript, where we discuss the limitations of our study.
  2. As suggested, the revised manuscript contains in the supplementary data additional pairwise comparisons between tumours.
  3. Reviewer 1 suggested that section 3.7 might be removed, as it ..”doesn’t bring much extra information”. We have followed this advice, mindful that reviewer 3 considered the manuscript …”a bit lengthy in parts”.

Minor issues.

  1. As suggested, the colours in the heatmaps have been changed for greater contrast. This is a great improvement.
  2. As suggested, references have been provided for ChIP-seq using antibodies against pol III.

Reviewer 2 Report

In their manuscript, Butterfield and colleagues examined the level of H3K27 acetylation in various tRNA genes in normal, tumour, and metastatic tissues of patients with ERα-positive luminal breast cancer. The authors postulate that the level of H3K27 acetylation directly correlates with the level of transcription of a particular tRNA and, therefore, with tRNA expression level. While there is literature available to support this claim, I would prefer to see a correlation analysis of H3K27 acetylation and transcription levels in breast cancer. The authors are well aware of the limitations of this approach, such as insensitivity to variations in tRNA stability caused by post-transcriptional modifications, but I would have preferred this to be more clearly stated in the Conclusion as well.

This paper is a good example of a research work based solely on data mining. Although its significance is moderate and the work represents immediate interest for relatively narrow group of researches, it is still a well-conceived and well executed study, which publication may more significant impact that reviewer anticipates.

Major issues:

1. Study of a correlation between H3K27 acetylation and transcription of tRNA genes need to be performed. Ideally using the same samples or if it is impossible using other breast cancer samples.

2. Authors need to articulate the limitations of this approach in Conclusions.

Minor issues:

1. I would suggest using other colours for the heatmap because poor visibility of the shade of the green used in the paper and potential issues related to colour blindness.

2. In sentences starting on lane 288 "Because of what was reported" and ending on lane  291 "Figure 4A)." and further in the manuscript, the authors use the terms tRNA-iMet-CAT and tRNA-iMet-CAU. They should provide an explanation of these terms for the non-tRNA expert reader.

no comments

Author Response

Reviewer 2 considered this “.. a well-conceived and well executed study” and “…a good example of a research work based solely on data mining”. We are grateful for these comments and for the suggestions for improvement.

Major issues

  1. The reviewer suggested that we investigate “correlation between H3K27 acetylation and transcription of tRNA genes” … “Ideally using the same samples or if it is impossible using other breast cancer samples”. This is not possible, as RNA was not extracted from the tumours used for H3K27ac ChIP-seq. Furthermore, we are unable to find in the literature another suitable source of data from breast cancer samples, so we are obliged to extrapolate from other cell types. We think this should be acceptable, as previous tRNA abundance measurements were found to correlate well with H3K27ac in each tumour type tested, which ranged from primary prostate, colon and bladder tumours, to glioblastomas and lymphomas, as well as multiple cell lines derived from these (Gingold et al. 2014). We acknowledge the reviewer’s concern that the same correlation has not been proven in breast cancers, but there is a wealth of data showing that H3K27ac is associated with gene activity in many cell types and systems and no exceptions have been unearthed, as far as we are aware. Nevertheless, we have explicitly referred to this caveat in the revised manuscript.  
  2. As requested, we have discussed again in our conclusions the limitations of our approach.

Minor issues

As suggested, the colours in the heatmaps have been changed for greater contrast. This is a great improvement.

The terms tRNA-iMet-CAT and tRNA-iMet-CAU have been explained in the revised manuscript.

Reviewer 3 Report

This manuscript, by Butterfield et al, is an extended analysis of tRNA genes in breast cancers, focusing on a few tRNA genes. It is based on analysis of publicly available data sets. It is also an extended introduction and discussion, to some extent resembling a review of key facts, questions and outstanding issues in tRNA research. I found the manuscript easy to read, though a bit lengthy in parts. It provides several interesting points for further research and discussion. It is certainly of interest to the field.

1.While I do think that the use of H3K27ac is a bit risky since the evidence for its strict association with transcriptional output can be challenged (and has been challenged),  it is one approach among others. And in the  future we can probably used this study and to compare with others, serving as one reference point. The authors may have emphasized this limitation of the study a bit more, but it is there in the introduction section so it is nothing I demand to be stated more clearly.

2.Note, I think the supplementary file is the figure legends already in the main text? So in that sense it is not needed. Please double check.

3.In my PDF file, I cant really see the individual gene name in the figures, this refers to fig 1 and 3, which is understandable in such long lists. Yet it would be of interest if there is a possibility to publish these as high resolution, possible to zoom-in images where the gene name is readable. These can be published as supplemental images. This is just a suggestion.

Author Response

We are glad that reviewer 3 found our manuscript interesting and are grateful for his or her comments.

  1. We agree that H3K27ac cannot be equated with an absolute measure of transcriptional output, but believe that enough studies of tRNA genes have demonstrated good correlation in a variety of cell types to warrant its use as an acceptable proxy. The revised manuscript contains additional discussion of the limitations of our approach, which has been added to the conclusions section to emphasise the caveats described in the introduction.

  1. We apologize for any confusion in uploading files and believe this has been sorted out.

  1. It should be possible to zoom-in to read individual gene names online.

Round 2

Reviewer 1 Report

The authors have satisfactorily addressed my comments/suggestions.